# Emerging Insights into Sall4’s Role in Cardiac Regenerative Medicine

**DOI:** 10.3390/cells14030154

**Published:** 2025-01-21

**Authors:** Jianchang Yang

**Affiliations:** Michael E. DeBakey Department of Surgery, Baylor College of Medicine, One Baylor Plaza, Houston, TX 77030, USA; jianchay@bcm.edu

**Keywords:** cardiac fibroblast, partial reprogramming, myocardial infarction, protein interactions, epigenetic regulation, cardiogenic

## Abstract

Sall4 as a pivotal transcription factor has been extensively studied across diverse biological processes, including stem cell biology, embryonic development, hematopoiesis, tissue stem/progenitor maintenance, and the progression of various cancers. Recent research highlights Sall4’s emerging roles in modulating cardiac progenitors and cellular reprogramming, linking its functions to early heart development and regenerative medicine. These findings provide new insights into the critical functions of Sall4 in cardiobiology. This review explores Sall4’s complex molecular mechanisms and their implications for advancing cardiac regenerative medicine.

## 1. Introduction

Cardiac regeneration remains a critical challenge in addressing the global burden of heart disease, as the adult human heart has a very limited capacity for self-repair. Following ischemic injuries like myocardial infarction (MI), this limited regenerative capacity often results in fibrotic scar formation, progressive cardiac dysfunction, and eventual heart failure. Despite significant advancements in pharmacological and surgical interventions, current therapies remain insufficient to restore lost myocardial tissue or reverse the decline in cardiac function. This persistent unmet need has driven extensive research into regenerative strategies, including cell therapy, tissue engineering, and direct cardiac cellular reprogramming [1,2].

Central to these approaches are the critical roles of transcriptional networks and epigenetic regulators, which orchestrate gene expression programs essential for reshaping cellular identity and facilitating repair. Cardiac transcription factors such as Gata4, Mef2c, Nkx2.5, MyocD, and Tbx5 drive cardiomyocyte regeneration by activating key gene networks. Additionally, epigenetic modifiers reshape the chromatin landscape to complement this process. Particularly, histone methyltransferases and deacetylases dynamically regulate chromatin accessibility, while chromatin remodelers—such as the nucleosome remodeling deacetylase (NuRD) complex and the Polycomb repressive complex (PRC)—reposition nucleosomes to expose cardiogenic enhancers. DNA methyltransferases (DNMTs) and demethylases further fine-tune cardiogenic gene networks by modifying CpG islands. This synergistic interplay between transcriptional and epigenetic modulators not only orchestrates the intricate processes of cardiac development, but also offers powerful tools to unlock the regenerative potential of cardiac cells and restore functional myocardium.

Recent research highlights Sall4’s indispensable roles in cardiac biology. Originally known for maintaining cellular stemness, Sall4 is now increasingly recognized for its involvement in early heart development, cardiomyocyte proliferation, and regeneration. In cardiac fibroblasts, Sall4 reactivates developmental pathways, promotes cell fate transitions toward progenitor and cardiomyocyte-like states, and synergizes with other cardiac factors to enhance myocardial repair. Beyond these functions, Sall4 appears to support recovery by activating angiogenic pathways, reducing fibrosis, and promoting cardiomyocyte survival. Importantly, it achieves these reparative effects without inducing tumorigenesis, addressing a critical concern about its safety as a therapeutic target.

This review examines the molecular mechanisms underlying Sall4’s diverse functions. By integrating perspectives from stem cell biology, epigenetics, and cardiac research, we explore Sall4’s distinctive role in cardiac regeneration and its promise as a potential therapeutic target.

## 2. Sall4 as a Multifaceted Regulator in Embryonic Stem Cells, Development, and Tissue Progenitors

Sall4, the most extensively studied member of the Sall gene family, is a homolog of the *Drosophila* region-specific homeotic gene *spalt* (*sal*) [3,4]. Initially associated with Okihiro syndrome, a condition characterized by ocular abnormalities, as well as limb and heart malformations [5,6,7], Sall4 has since been recognized for its diverse and critical roles in embryonic stem cells (ESCs), organ development, somatic progenitor regulation, and oncogenesis [8,9,10]. These functions are mediated by its unique genetic structure, which contains multiple Cys2His2 zinc finger domains essential for DNA binding and protein–protein interactions. These interactions enable Sall4 as a potent activator or repressor, depending on specific cellular context and biological process [8,9,10].

In ESCs, Sall4 forms an autoregulatory network with Oct4, Sox2, and Nanog to maintain self-renewal and pluripotency. In extraembryonic endoderm (XEN) cells, however, Sall4 sustains an undifferentiated state by modulating lineage-associated genes, such as *Gata4*, *Gata6*, *Sox7*, and *Sox17*. Moreover, these cell type-dependent regulatory cascades involve mutually exclusive H3K4me3 activation marks and H3K27me3 repressive marks that regulate gene expression [11]. In murine postgastrulation embryos, Sall4 regulates neuromesodermal progenitor (NMP) differentiation, with its binding sites specifically enriched in neural and mesodermal genes. Mutations in Sall4 result in neural tissue expansion, mesodermal tissue reduction, and body axis truncation [12,13]. In zebrafish embryos, Sall4 interacts with the homeobox protein CDX-4 to coregulate genes essential for mesoderm-to-blood transitions. Mutations in both genes result in a reduced number of hemoglobinized red blood cells [14]. During limb development, a Sall4-Gli3 regulatory system is detected in limb progenitors, coordinating the formation of skeletal elements [15]. In the embryonic liver, Sall4 is selectively expressed in hepatic stem/progenitor cells (hepatoblasts), where it inhibits their maturation into hepatocytes while promoting cholangiocyte formation [16].

Sall4 also plays a role in germline development, ensuring proper primordial germ cell (PGC) specification and supporting postnatal spermatogonial progenitor cell (SPC) differentiation [17,18]. In neonatal human hearts (1 day–1 month), Sall4 is expressed in Islet-1(Isl1)+ cardiac progenitor cells (CPCs), along with SOX2, EpCAM, and TBX5, to modulate their stemness features [19]. This parallels the Sall4 expression in the progenitors and cardiomyocytes of the second heart field (SHF) during mouse heart development, where Sall4 forms a complex with Myocd and SRF to regulate cell proliferation [20].

In human bone marrow, SALL4 is expressed in CD34+ hematopoietic stem/progenitor cells (HSPCs), where it regulates genes essential for self-renewal and hematopoietic differentiation, such as CD34, HOXA9, RUNX1, and PTEN [21]. Furthermore, overexpression of SALL4 stimulates the expansion of HSPCs both in culture and in serial transplantation models in vivo [22,23]. These findings underscore its diverse regulatory roles across developmental contexts and highlight its therapeutic potential.

Unsurprisingly, SALL4 can become oncogenic when dysregulated. Aberrant expression of SALL4 has been identified in over 10 types of solid tumors and several forms of leukemia, where it drives stem-like traits, promoting cellular proliferation, survival, and metastatic potential. This positions SALL4 as a key biomarker and a promising therapeutic target for specific tumors and leukemias [8,9,10,24,25]. Collectively, SALL4 exemplifies remarkable regulatory versatility, offering valuable insights into both developmental biology and cancer therapies.

## 3. Sall4’s Role in Heart Development and Regeneration

Murine gene mutation studies have provided profound insights into Sall4’s role in heart development. In developing mouse embryos, Sall4 expression has been detected in ventricular myocardium at E10.5, with elevated levels in the interventricular septum, persisting through E12.5 [26,27]. Homozygous Sall4 knockout mice fail to survive beyond early embryonic stages, whereas heterozygous *Sall4* disruption results in a range of abnormalities affecting the anorectal, limb, and heart regions. The heart defects include abnormal interventricular groove formation and myocardial disorganization, which overlap with phenotypes caused by Gata4 and Tbx5 mutations [28,29] and model several genetic syndromes seen in human patients [30,31,32]. In another model, overexpression of a truncated form of Sall4 (ΔSall4), which inhibits both Sall4 and Sall1 function, results in a hypoplastic right ventricle, outflow tract defects, disrupted ventricular septum formation, and thinner ventricular walls, along with decreased proliferation of second heart field (SHF) progenitors and cardiomyocytes [20]. In these studies, critical regulatory partners and downstream genes of Sall4 have been identified, including its homolog Sall1, T-box family member Tbx5, MyocD, serum response factor (SRF), Isl1, Gap junction alpha-5 (Gja5), cyclin genes and cyclin-dependent kinases (CDKs). These factors play diverse roles in mesodermal differentiation, cardiac progenitor development, myocardial lineage commitment and proliferation, underscoring their mechanistic interactions with Sall4 in these processes.

## 4. Sall4’s Role in Cardiac Cellular Reprogramming and Therapeutics

The rapid development in developmental pathways has significantly propelled research in cardiac regenerative medicine, particularly in the field of cardiomyocyte reprogramming. In the early 2010s, several landmark studies revealed that overexpression of specific transcription factors, such as Gata4, Mef2c, and Tbx5 (GMT), can directly transdifferentiate cardiac fibroblasts into induced cardiomyocyte-like cells (iCMS) both in vitro and in animal MI models [33,34,35,36,37]. This groundbreaking approach opened a new avenue in regenerating injured heart tissues and enhancing functional recovery. However, subsequent studies encountered challenges, including low efficiency and inconsistency in generating mature and functional iCMs, particularly in higher-order species like pigs and humans [38,39,40]. These limitations significantly hamper its therapeutic applications, necessitating the development of improved reprogramming factors or innovative methodologies to fully realize their application potential.

Our group has worked on Sall4 due to its direct interactions with the iCM reprogramming factors Tbx5 and Gata4. We initially substituted Gata4, Mef2c, or Tbx5 individually with Sall4 in both mice and rat cardiac fibroblasts. While overexpression of GMT upregulated the mRNA levels of stemness factors such as *Sall4*, *Bmi1*, and *p63*, the combination of Gata4/Mef2C/Sall4 triggered a remarkable increase in *Oct4* and *Nanog* mRNA expression over seven days. Prolonged culture led to the formation of stem-like clusters that were partially positive for alkaline phosphatases staining [41]. In extended studies, the combination of Sall4 and Gata4, which physically interact, stimulated cardiac fibroblast transitions into partially stem/progenitor-like cells with enhanced ex vivo expandability and clonogenic potential [42]. Moreover, these reprogrammed cells demonstrated strong cardiogenic potential, differentiating into contractile cardiomyocytes, endothelial cells, and even neuron-like cells. Notably, this cardiogenic reprogramming process also showed efficacy in human cardiac fibroblasts, highlighting its potential for clinical applications (Table 1) [42,43].

In another study, Zhao et al. screened 22 candidate transcription factors predicted through the analysis of gene regulatory networks in cardiac development [44]. They found that five factors, GMTMS (GMT plus Myocd and Sall4), induced significantly more iCMs expressing the cardiac structural proteins cTnT and cTnI than GMT alone in fibroblasts isolated from both normal and infarcted mouse hearts. Furthermore, GMTMS induced abundant beating cardiomyocytes by day 28 postinfection. Mechanistically, Myocd contributed mainly to inducing the expression of cardiac proteins, while Sall4 accounted for the induction of functional properties, such as contractility (Table 1). Although the detailed molecular mechanisms and specific reprogramming consequences remain to be fully elucidated, these studies highlight the critical potential of Sall4 applications in driving myocardial repair.

## 5. Sall4-Mediated Regulatory Networks and Pathways

A comprehensive understanding of Sall4’s regulatory mechanisms is essential for advancing its regenerative functions and therapeutic potential. These include its key interacting transcriptional networks, epigenetic partners, and critical downstream signaling pathways involved in cardiac biology (Figure 1) [26,27,45,46,47,48,49]. In the following sections, we classify and explore these aspects to lay the groundwork for future advances in regenerative medicine and targeted cardiac therapies.

### 5.1. Interacting Transcription Factors (TFs)

Sall1: Sall1 shares functional similarities with Sall4 and compensates for it in maintaining cardiac progenitors, regulating lineage specification, and directing heart patterning [26,50,51,52,53,54,55]. Both factors exhibit genetic interactions and overlapping expression patterns in the developing murine heart, and disruption of both genes results in more pronounced heart defects [20,27]. Beyond this, Sall1 and Sall4 redundantly influence other organs, including neural patterning, differentiation, and morphogenesis, as observed in both murine and Xenopus laevis models [56,57]. In reprogramming studies, Sall1 cooperates with Sall4 and other factors to convert embryonic fibroblasts into induced pluripotent stem cells (iPSCs) without requiring Oct4 [58,59]. In cardiac fibroblasts overexpressing Sall4 and Gata4, Sall1 expression is significantly increased, facilitating cellular fate transition to a stem-like state [42]. These findings underscore the importance of exploring the cooperation between Sall1 and Sall4 to enhance cardiac reprogramming and advance regenerative strategies.

Gata4: Gata4 is a master regulator in both heart development and cardiac regeneration, playing a central role in GMT-induced iCM reprogramming. Its high endogenous expression in cardiac fibroblasts, like Tbx20, allows even modest levels of Gata4 to effectively drive iCM generation [60,61]. When coexpressed with Sall4, the two factors synergistically promote the reprogramming of cardiac fibroblasts, activating key pluripotency and cardiogenic genes, including Oct4, Lin28, Nkx2.5, Tbx5, Sox17, FGF10, Isl1, and SRF, thus enhancing the regenerative potential of these cells [42]. Remarkably, all GATA family members can induce pluripotency in the absence of Oct4. In this process, both Gata4 and Gata6 directly target Sall4, which acts as a bridge linking the pluripotency network [62]. Additionally, like Sall4, Gata4 has been shown to promote cardiac differentiation from embryoid bodies (EBs) by activating the endodermal gene *Sox17* [63,64,65,66]. This activation, in turn, can facilitate cardiomyogenesis in a noncell-autonomous manner through the endodermal cardiogenic factor Hex [63,67]. Thus, the Sall4/Gata4/Sox17/Hex cascade may represent a central pathway driving cardiogenic differentiation in Sall4/Gata4-mediated reprogramming.

Oct4 and Sox17: Whole-mount immunostaining in mouse embryos (E7.5) has revealed the coexpression of Sox17 with both Sall4 and Oct4 in extraembryonic visceral endoderm and mesendoderm, suggesting their coordinated role in directing cell transitions from the epiblast towards the cardiogenic mesoderm [68]. On the other hand, Sall4 has been shown to mediate Oct4’s switching from the *Sox2* to the *Sox17* promoter, accompanied by their changes in H3K27me3 and H3K4me3 histone markers, respectively. Sall4 also recruits PRC1/PRC2 complexes to *Sox2* locus, and the PRC1 complex to *Sox17* locus. Thus, this interplay acts as a molecular switch, transitioning Oct4/Sox2-driven pluripotency to Oct4/Sox17-mediated cardiac specification [69]. Consistently, Sall4 and Gata4 overexpression in cardiac fibroblasts led to robust Sox17 expression at both mRNA and protein levels [42]. Therefore, future investigations should delve into these related molecular switching mechanisms to enhance cardiac regenerative strategies.

Tbx5: Tbx5 is another critical Sall4-interacting factor in cardiac development. Studies have shown that these two factors cooperate both positively and negatively to fine-tune downstream gene transcription, orchestrating precise murine heart patterning and morphogenesis [70,71,72,73]. A recent study further identified a unipotent Tbx5+ embryonic cardiac precursor population, which can differentiate into cardiomyocytes and is also detected in the infarcted adult murine heart [73]. In addition to this, Tbx5 is also expressed in cardiac fibroblasts [60], where it acts as a central factor in iCM reprogramming to compensate for heart cell loss. Furthermore, during Sall4/Gata4-mediated cardiac reprogramming, Tbx5 emerged as one of the most upregulated genes [42], further highlighting its critical role in enhancing cardiac regeneration and repair.

Myocd and SRF: Myocd is a transcriptional coactivator that does not bind directly to DNA but induces gene expression through interactions with SRF. Myocd is required for cardiac specification, cardiomyocyte survival, and maintenance of heart function [74,75], while SRF, a cardiac-enriched TF, is crucial for sarcomere formation [76]. In embryo-derived trabecular and compact layer cardiomyocytes, Sall4 has been found to form endogenous complexes with both Myocd and SRF, synergistically regulating cell cycle-related genes, promoting cardiomyocyte proliferation and advancing heart regeneration [20]. During iCM reprogramming, incorporating Myocd with Sall4 or SRF into the GMT cocktail enhanced the efficiency in both rodent and human cell systems [44,77,78,79]. Notably, Sall4/Gata4-mediated reprogramming in cardiac fibroblasts significantly upregulated SRF but not Myocd, suggesting a distinct regulatory role for SRF [42]. Given the competitive protein interactions and functional dynamics among Sall4, pluripotency networks, and essential cardiogenic factors, strategically incorporating Myocd and SRF into the Sall4/Gata4 complex may further drive cardiogenic identity and functionality.

Isl1: Isl1 plays a critical role in regulating CPC differentiation and fate determination during early heart development. In neonatal patient-derived CPC clones, a population of Isl1+ cells has been identified coexpressing Sall4, Sox2, EpCAM, and Tbx5, with Sall4 and TFAP2C marking an early premesendoderm stage [19]. Furthermore, Isl1 regulates CPC differentiation through interactions with Notch1 and β-catenin signaling [80]. Sall4 also regulates cell proliferation by activating Isl1 in SHF progenitors and cardiomyocytes, which is crucial for CPC development. In the anterior heart field (AHF), Isl1 and GATA factors coregulate Mef2c transcription. Isl1 further interacts with Nkx2.5 to control its cardiac localization and cardiomyocyte cell fate [81]. Additionally, Isl1 induces the differentiation of mesenchymal stem cells (MSCs) into cardiomyocyte-like cells through enhanced binding of histone acetyltransferase Gcn5 to the promoters of *Gata4* and *Nkx2.5*, while reducing DNMT1 binding at *Gata4* promoter [82]. Consistently, in Sall4/Gata4-reprogrammed cardiac fibroblasts, the upregulation of Isl1 further highlights its significance in cardiac progenitor development and regeneration.

### 5.2. Epigenetic Regulation Partners

Epigenetic modifying machineries critically influence Sall4’s regulatory functions, many of which are vital in regenerative cardiac biology [8,9,10,83,84]. Key examples include DNMTs, histone modifying enzymes [85,86,87], and several chromatin-modifying complexes. Additional contributors, such as the Set histone methyltransferase Whsc1, further highlight the complexity of Sall4’s epigenetic interactions.

DNMTs: All central DNMTs, including DNMT1, DNMT3a, DNMT3b, and DNMT3l, critically modulate cardiogenic lineage commitment, embryonic cardiomyocyte differentiation, maturation, contractile function, and the pathogenesis of cardiomyopathies and cardiac fibrosis [88,89,90,91,92]. DNA methylation (DNAm) biomarkers and several DNMTs are being explored as potential therapeutic targets for cardiovascular diseases, including cardiac fibrosis, ischemic injury, myocardial infarction, and heart failure progression, with a focus on modulating their activity to correct aberrant DNA methylation patterns [92,93,94,95,96]. Given that Sall4 recruits all these DNMTs, and that Sall4/Gata4-mediated cardiac reprogramming involves heightened levels of DNMT3b and DNMT3l [8,42,97], it will be essential to monitor specific DNMT activity and targeted methylation patterns in genes during the reprogramming process. These insights are crucial to understanding their roles in cardiac regeneration and therapeutic applications.

HDAC1/2: Knockout studies have shown that HDAC1 and HDAC2 redundantly control myocardial morphogenesis, growth, and contractility in the embryonic heart [98]. In diseased models, targeted inhibition of HDAC1/2 or all classical HDACs has been shown to inhibit cardiac hypertrophy, protect against oxidative damage, inhibit inflammation, inhibit fibrosis, and modulate the extracellular matrix (ECM) composition [99,100]. Furthermore, HDAC inhibition enhances direct cardiac cellular reprogramming by epigenetically reactivating genes involved in cardiac fate acquisition [79,101,102,103]. In both development and disease models, HDAC1 and HDAC2 interact with Sall4 to repress downstream gene expression, such as *Sall1* and *Pten* [8,9,86,104]. Increased HDAC1 activity is also observed during Gata4/Sall4-mediated cardiogenic reprogramming and differentiation [42], reinforcing its dynamically regulated role in cardiac fate transitions and function maintenance.

Lsd1: Lsd1 (lysine-specific demethylase 1) is essential for murine embryonic and neonatal heart development. It regulates cardiomyocyte proliferation by repressing noncardiac genes, such as the neuron-specific gene *Cend1*, through the removal of H3K4me2 marks at its promoter [105]. Consequently, downregulation of Lsd1 and its demethylation activity impairs neonatal heart regeneration [106]. This Lsd1-Cend1 axis also critically controls the proliferation of human iPSC-derived cardiomyocytes (hiPSC-CMs). Additionally, cardiac-specific Lsd1 inactivation in mouse models results in myocardial hypertrophy and heart failure. In ischemia/reperfusion (I/R) injury and oxygen–glucose deprivation (OGD)-induced injury, Lsd1 expression decreases, leading to increased histone methylation. However, Lsd1 overexpression improves the viability of fibroblasts, reduces apoptosis, and diminishes reactive oxygen species (ROS) production after injury [107]. In cardiac fibroblasts, Lsd1 regulates important processes such as cell survival and reprogramming. Lsd1 inhibition combined with forskolin (adenylyl cyclase activator) significantly enhances iCM reprogramming efficiency, underscoring its importance in regenerative medicine [108]. Furthermore, Lsd1 interacts with DNMT1, regulating heart defects through the control of E-cadherin phosphorylation [109]. Lsd1 and DNMT1 are also involved in Sall4-mediated transcriptional repression in hematopoietic stem cells [8,9,110]. Importantly, Lsd1-targeted therapy has shown promise in preventing cardiomyopathy in laminopathy mouse models [111,112], suggesting a potential strategy for therapeutic intervention in heart disease.

Dot1l: Dot1l epigenetically catalyzes the methylation of histone H3 at lysine 79 (H3K79me1/2/3), facilitating gene activation through interactions with coactivators or TFs such as Sall4, Oct4, cMyc, and Nkx2.5 in various cellular contexts. In human ESCs, Dot1l activation significantly promotes cardiac lineage differentiation programs [113]. During cardiogenesis, Dot1l-mediated H3K79me2 modification regulates gene expression critical for cardiomyocyte differentiation and orchestrates chamber-specific transcriptional programs, while also mediating postnatal cardiomyocyte cell cycle withdrawal [114,115]. Additionally, Dot1l regulates *dystrophin* gene expression, which is crucial for cardiac contractile function [116]. Notably, inhibition of Dot1l, particularly in cardiac fibroblasts, prevents cardiac fibrosis and dysfunction, positioning it as a promising therapeutic target for remodeling cardiovascular diseases [117].

PRC1/2: Both PRC1 and PRC2 complexes are essential chromatin modifiers in orchestrating normal heart development [118]. Bmi1, a key PRC1 subunit, represses gene transcription through the ubiquitylation of histone H2A at lysine 119 [119]. In hematopoietic stem cells, Sall4 binds to the *Bmi1* promoter and regulates hematopoiesis through dose-dependent repression [120,121]. Like Sall4, Isl1, and Sca1, Bmi1 is critical for maintaining the primitive properties of CPCs, making Bmi1+ CPCs a promising target for advancing myocardial repair strategies [122]. In cardiac fibroblasts, Bmi1 is highly expressed and epigenetically inhibits the reprogramming of fully differentiated iCMs [122]. Furthermore, Bmi1 promotes cardiac fibrosis in ischemia-induced heart failure via the PTEN-PI3K/Akt-mTOR signaling pathway [123]. These diverse roles of Bmi1 highlight its importance not only in heart development but also in disease contexts and regenerative medicine. PRC2 also orchestrates developmental processes through its subunits, such the histone methyltransferase EZH2. Inactivation of EZH2 leads to severe congenital heart malformations, accompanied by the dysregulation of pivotal genes such as Cdkn2a, Isl1, Pax6, Myh6, Six1, and Hcn4 [124]. The dual regulation of Sall4 by EZH2 and KDM6A, as observed in tumor progression via the Wnt/β–catenin pathway [25,125], may suggest parallel epigenetic mechanisms in the cardiac context that warrant further investigation. These findings emphasize the intricate roles of Polycomb-group proteins in both health and disease contexts.

NuRD: The nucleosome remodeling and deacetylase (NuRD) complex governs biological processes through precise interactions with specific cofactors. During heart development, CHD4, a core NuRD component, interacts with cardiac TFs such as Gata4, Nkx2-5, and Tbx5 to coordinate cardiac lineage specification. By recruiting CHD4, Gata4 and Nkx2-5 repress noncardiac gene programs, ensuring the fidelity of cardiac specification [126]. Similarly, NuRD collaborates with Tbx5 both biochemically and genetically to suppress inappropriate gene programs that could disrupt cardiac development—a mechanism critical for cardiac septation—as revealed by the Tbx5 cardiac interactome [127]. Beyond heart development, NuRD partners with Sall4 to silence somatic chromatin loci in the early phase of iPSC reprogramming, underscoring its adaptability in orchestrating lineage-specific chromatin states [86,104]. Furthermore, CHD4 and the NuRD complex directly regulate cardiac sarcomere formation, highlighting their indispensable role in maintaining proper cardiac structure and function [128].

Whsc1: Whsc1 (also known as NSD2) is a histone methyltransferase family member that regulates H3K36 me2/me3, an activation mark on euchromatin [129]. In ESCs, Whsc1 specifically interacts with Sall1, Sall4, and Nanog, while in embryonic hearts it associates with cardiac TFs such as Nkx2-5. Whsc1-deficient mice exhibit growth retardation and midline defects, including congenital cardiovascular anomalies resembling Wolf–Hirschhorn Syndrome (WHS). The impact of Whsc1 haploinsufficiency is further exacerbated in Nkx2-5 heterozygous mutant hearts [130], highlighting the important functional links between these factors during cardiac development.

### 5.3. Signaling Pathways

In line with the complex transcriptional and epigenetic interactions described above, a range of Sall4-modulated signaling pathways is involved in cardiobiology, with several important pathways outlined below.

Wnt/β-catenin: This pathway is critically involved in mesodermal and CPC specification and early cardiomyocyte proliferation [131,132]. During embryogenesis, it exhibits biphasic, stage-dependent effects [133], and constitutive β-catenin activation in Isl1 lineage cells leads to Isl1+ cell accumulation in SHF-derived structures, including the right ventricle and outflow tract [134,135,136]. After birth, β-catenin promotes the hypertrophic growth of cardiomyocyte, with activity diminishing as the heart matures. Under pathological conditions, acute short-term β-catenin activation has been shown cardioprotective, whereas chronic activation contributes to hypertrophy and heart failure. In myocardial infarction models, exogenous β-catenin administration reduces infarct size, demonstrating antiapoptotic effects and cell cycle activation in both cardiomyocytes and myofibroblasts [137,138]. Sall4 enhances this pathway by directly binding to the *CTNNB1* gene promoter to increase β-catenin expression and upregulates key signaling components such as Wnt3a, Bmi-1, and downstream targets including c-Myc and CCND1, supporting the balance between progenitor self-renewal and differentiation [8,9,10]. Although Sall4’s role in promoting Wnt/β-catenin signaling is well documented in tissue stem-like cells, including in various cancers, its interactions within cardiac biology remain underexplored, presenting a critical area for future research in cardiac regeneration and therapy.

PI3K/AKT: In Sall4/Gata4-mediated partial stem/progenitor-like reprogramming, the PI3K/AKT pathway emerged as the most enriched, with 73 differentially expressed genes involved [42]. This pathway enhances the conversion of partially reprogrammed iCMs into mature cardiomyocytes, alongside the involvement of IGF1, mTORC1, and Foxo3a pathways [139]. It also induces cardiomyocyte dedifferentiation and cell cycle activation in human ESC-derived CPCs, resembling a rejuvenation process [140]. Dual inhibition of PI3K/AKT and MAPK pathways further promotes the maturation of hiPSC-CMs, leading to hypertrophy, multinucleation, improved calcium handling, and enhanced electrophysiological properties [141]. Beyond these roles, this pathway is critically involved in heart diseases, regulating cardiomyocyte size, survival, angiogenesis, inflammatory responses, and various cardiac disorders [142,143]. It also plays a key role in cardiac fibrosis by modulating cell survival, apoptosis, cardiac contractility, and the transcription of fibrosis-related genes [144]. Notably, SALL4 expression has been linked to proliferative, invasive, and antiapoptotic effects through activation of the PI3K/AKT pathway in various cancer stem-like cells [145,146]. These findings suggest a shared mechanistic interaction between Sall4 and the PI3K/AKT pathway in cardiac regeneration and disease, potentially representing a promising avenue for advancing therapeutic strategies in cardiac repair.

Retinoic acid (RA): RA signaling plays a diverse, stage- and dosage-dependent role in cardiac development, encompassing mesoderm formation, patterning of vertebrate cardiac progenitor fields, cardiomyocyte specification, epicardium development, outflow tract formation, ventricular compact wall growth, and coronary vasculature development [147,148,149]. In vitro, RA application induces the differentiation of various subtypes of cardiovascular cells from a pluripotent state, with outcomes determined by specific dosages and protocols [150,151]. Notably, RA receptors RXR and RAR bind to the same regulatory region of Sall4, mediating all-trans retinoic acid (ATRA)-dependent expression of the Sall4 isoform and maintaining an immature fate in germ cells [152,153]. Conversely, Sall4 directly binds to the promoter of the RA receptor *RARα* in leukemic cells, modulating target gene expression through the recruitment of Lsd1 [154]. Knockdown of Sall4 enhances ATRA-induced differentiation [154], suggesting a feedback mechanism between RA signaling and Sall4-mediated transcriptional regulation. These insights likewise warrant further investigation in the context of cardiac regenerative biology.

TGF-β: Sall4 has been shown to enhance TGF-β1 expression by binding to its promoter and regulate epithelial-to-mesenchymal transition (EMT)-related transcription factors such as Snail, Slug, Twist, and ZEB1 [155,156]. The TGF-β/Smad signaling plays a pivotal role in cardiac fibrosis by activating cardiac fibroblasts and accelerating extracellular matrix (ECM) production in injured tissues [157]. Suppression of this signaling allows GATA4 to interact with the H3K27me3 demethylase JMJD3, facilitating the reversal of silencing marks and promoting the transcription of cardiogenic genes [158]. During cardiac reprogramming, inhibiting TGF-β signaling or combining it with WNT inhibitors has been shown to significantly enhance GMT-induced iCMs both in vitro and in vivo. In Sall4/Gata4-mediated partial cardiac reprogramming, this pathway is among the most enriched [42], suggesting a critical role in mediating cardiogenic cellular fate conversion.

Taken together, Sall4 exerts its diverse regulatory roles through complex, dynamic interactions with various TFs, epigenetic modifying partners, and signaling pathways, many of which are vital in heart development, diseases, and cardiac regeneration and repair (Figure 1). A deeper understanding of these critical interactions is expected to provide valuable mechanistic insights into Sall4’s role and significantly advance the development of novel or improved targeted cardiac therapies.

### 5.4. Competitive and Dynamic Regulation of Interacting Pathways

As discussed, dynamic and competitive interactions among protein complexes, including tissue-specific and stage-dependent induction or repression of TFs, epigenetic modifiers and chromatin remodelers, and signaling pathways, profoundly influence cellular identity and function [159]. These interactions are equally pivotal in cellular reprogramming, where outcomes depend heavily on the overexpressed factors and the cellular source. For example, Gata4, a versatile transcription factor, can reprogram cells into various phenotypes, such as iPSCs, CPCs, or iCMs, depending on the cocktail of factors used and the target cell type (Figure 2). Specifically, while both Gata4 and Sall4 can induce pluripotency, their combined overexpression uniquely generates cardiogenic stem/progenitor cells in cardiac fibroblasts but not in skin fibroblasts [42]. This phenomenon may be attributed to (1) Sall4′s intrinsic role in recruiting pluripotency-related networks and epigenetic regulators, (2) Gata4′s inherent cardiogenic activity and its activated signaling pathways, and (3) the strong cardiogenic expression profiling and potential of cardiac fibroblasts. These cell type-dependent effects offer distinct advantages for therapeutic applications. Conversely, the inclusion of alternative, mutually interacting factors, such as Tbx5, Nkx2.5, Mesp1, Baf60 [160], or Tbx5, Mef2, and Myocd [44,160], restricts reprogramming to an induced cardiac progenitor (iCPC) or committed cardiomyocyte phenotype (iCM) by inhibiting the pluripotency pathway and endogenous interactions. Thus, the specific combination of factors used in reprogramming dictates the resulting cell type and phenotype achieved, tailoring each approach for specific therapeutic interventions.

## 6. Therapeutic Potential for Sall4 in Cardiac Regeneration and Repair

Emerging research underscores Sall4 as a promising candidate for enhancing cardiac regeneration. In a myocardial infarction (MI) model, administration of Sall4 has demonstrated reparative effects, including its ability to promote the transdifferentiation of myofibroblast into cardiomyocyte-like cells, in conjunction with GMT, Myocd, and two chemical compounds [161]. This process was associated with improved left ventricular ejection fraction and fractional shortening, observed six weeks post-MI and treatment [161]. In our study, the combination of Gata4 and Sall4 exerts synergistic effects, facilitating the conversion of cardiac fibroblasts into progenitors and lineage cells essential for cardiac repair [42]. This strategy provides a robust platform for investigating cardiac regeneration and evaluating potential therapeutic strategies [43]. Furthermore, consistent with the reparative effects of Gata4, our experimental data suggest that Sall4 contributes to critical aspects of the cardiac healing process, including promoting cell survival, reducing fibrosis, and stimulating angiogenesis [41,42,43,162]. These processes are pivotal for restoring heart function after injury.

As a potential single-agent or combined therapy, Sall4 holds significant therapeutic promise, especially with the identification of new mechanisms and enhancers in the field. However, its safety and efficacy must be carefully evaluated, particularly with respect to potential long-term outcomes. In this regard, although dysregulated Sall4 expression has been implicated in various tumors, the risk of tumorigenicity in cardiac disease treatment appears negligible. Sall4 administration has been applied in MI animal studies, with no such effects documented [41,161]. In xenotransplant models, both of the two Sall4 isoforms—Sall4a (full-length) and Sall4b (alternatively spliced)—expanded hematopoietic cells, supported long-term engraftment, and did not disrupt niche regulation or lead to tumor formation, supporting their utility in achieving clinically significant expansion of human HSCs [22,23]. Additionally, our group previously generated conditional Sall4b knock-in mice targeting the cardiac fibroblast fraction (controlled by a *Col1a1* promoter), with no tumors observed throughout their lifetimes (author’s unpublished data). Although the Sall4b transgenic mice, driven by a CMV promoter, exhibited leukemic phenotypes at early stages, this was likely attributed to Sall4b dysregulation within multiple tissue systems during the embryonic period [163]. In contrast, all Sall4a transgenic mice remained healthy and tumor-free throughout their lifetime. Collectively, these findings underscore the potential applicability of Sall4 isoforms in the context of cardiac therapy.

## 7. Conclusions

In summary, recent studies highlight the potential of Sall4-based interventions as a promising therapeutic strategy for cardiac regeneration. Sall4’s ability to drive cell fate transitions and enhance reparative processes positions it as a key player in regenerative medicine. Its synergistic interactions with pivotal cardiac factors such as Gata4, Tbx5, Nkx2.5, Myocd, and Isl1, coupled with its recruitment of essential epigenetic regulators and modulation of critical signaling pathways, further emphasize its potential in combination therapies for optimizing cardiac repair. However, the successful clinical translation of Sall4-based strategies depends on a more comprehensive understanding of its molecular mechanisms, the development of precise targeting methods, and a thorough evaluation of long-term efficacy and safety profiles. Continued research into novel regulatory pathways and enhancers will be essential to refine these therapeutic approaches, establishing the way for advancements in cardiac regenerative medicine.

## Figures and Tables

**Figure 1 cells-14-00154-f001:**
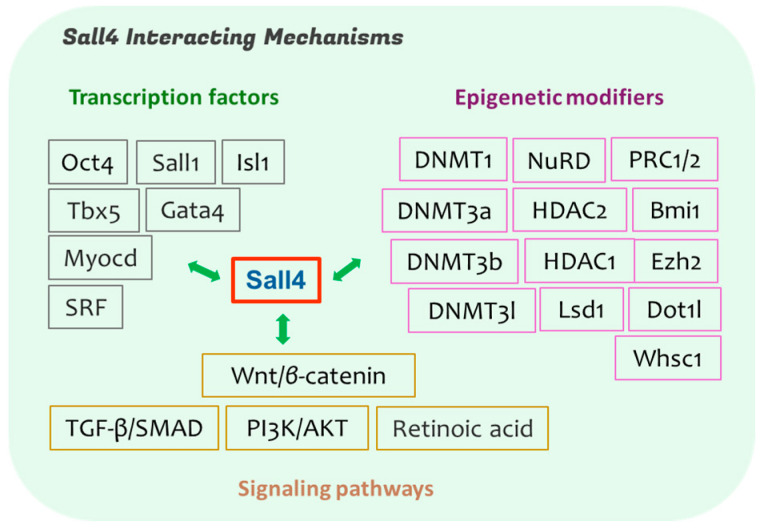
Sall4’s regulatory networks involved in cardiac development and regeneration. These networks encompass key stemness and cardiogenic transcription factors, chromatin modifiers and epigenetic regulators, and multiple signaling pathways. Together, they play a pivotal role in regulating stem cell identity, cardiac specification, functional development, and cellular reprogramming. These interconnected pathways provide a strong foundation for advancing targeted therapies to enhance cardiac repair and regeneration.

**Figure 2 cells-14-00154-f002:**
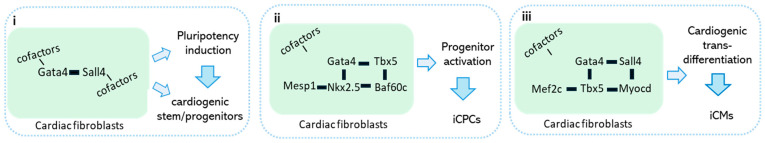
Competitive reprogramming pathways drive cardiac cell fate outcomes. Overexpression of Sall4 or Gata4 in cardiac fibroblasts can lead to the generation of distinct cell types, including induced cardiogenic stem/progenitors (**i**), cardiac progenitors (iCPCs) (**ii**), or cardiomyocyte-like cells (iCMs) (**iii**). These outcomes are shaped by dynamic and competitive genetic interactions within the reprogramming cocktails, along with the recruitment of specific endogenous coregulators.

**Table 1 cells-14-00154-t001:** Studies exploring Sall4’s role in cardiac reprogramming and human implications.

Study/ Reference	Model System	Approach	Key Findings	Impact on Human Cells
Initial GMT Studies; [41]	Cardiac fibroblasts from neonatal and adult mice and rats	Overexpression of Sall4 substituted for Gata4, Mef2c, or Tbx5 in GMT reprogramming factors	Sall4 combined with Gata4/Mef2c upregulated stemness genes (e.g., Oct4, Nanog); stem-like clusters formed with positive AP staining	Showed partial reprogramming potential in rodent models
iCM Reprogramming; [44]	Mouse cardiac fibroblasts	Analysis of regulatory networks; GMT + Myocd and Sall4 (GMTMS) identified as optimal combination	GMTMS induced more cardiomyocyte-like cells expressing cTnT and cTnI and functional beating cardiomyocytes by day 28	Sall4 linked to functional properties such as contractility; potential translational applications
Sall4 + Gata4 Studies; [42]	Mouse and rat cardiac fibroblasts	Sall4/Gata4-based approach; physical interaction enhanced cardiac fibroblast fate transition	Partially stem/progenitor-like cells with clonogenic potential; differentiated into cardiomyocytes, endothelial cells, and neuron-like cells	Partial reprogramming established in rodent cardiac fibroblasts (clinical potential)
Extended Human Cell Studies; [42]	Human cardiac fibroblasts	Sall4 combined with Gata4	Strong reprogramming capacity: generating contractile cardiomyocytes and enhanced cardiogenic potential	Demonstrated efficacy in human cells, highlighting therapeutic relevance.

## Data Availability

Not applicable.

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
