# Peer review of "Emerging Insights into Sall4’s Role in Cardiac Regenerative Medicine"

_cells, 2025, doi:10.3390/cells14030154_

Round 1

Reviewer 1 Report

Comments and Suggestions for Authors

This is a very nice review article about the role of Sall4 in cardiac regeneration. This is indeed and interesting and emerging factor which regulates pluripotency, induced cardiogenic stem cell or progenitor formation. Also, in the so-called transdeferentiation of fibroblasts into cardiomyocytes. The review is very well written, timely and novel. I have the follwing suggetsions:

1. Formatting issue at the begining of the sections 5.2 and 5.3 should be fixed.

2. Line 383- the abbreviation ATRA should be explained at first mention. 

3. Figure 1 could be improved with additional information. I would have included maybe a few bullet points or keywords explaning the major functional effects of each signaling pathway (from the text)

4. Conceptionally, lots of information of given about the effects of Sall4 overexpression in cell cultures, e.g. fibroblasts etc. As a reader I would be more interested to know about the effects of Sall4 overexpression in mice. At the end, lines 460-468 some of this imformation is mentioned, also mentioning Sall4a and Sall4b. Are these twoisoforms of Sall4?? This should be clearly explained. What is the difference between them. Does the unpublished knockin mouse cause overexpression of Sall4a or Sall4b in all or paticular tissues?  

Author Response

Reviewer 1

This is a very nice review article about the role of Sall4 in cardiac regeneration. This is indeed and interesting and emerging factor which regulates pluripotency, induced cardiogenic stem cell or progenitor formation. Also, in the so-called transdeferentiation of fibroblasts into cardiomyocytes. The review is very well written, timely and novel. I have the follwing suggetsions:

  1. Formatting issue at the begining of the sections 5.2 and 5.3 should be fixed.

Response: These formatting issues have been addressed. Please see the revised text in lines 249–253 and 350–352.

  1. Line 383- the abbreviation ATRA should be explained at first mention. 

Response: We have provided an explanation for ATRA at its first mention in the text (see yellow-highlighted words in line 394).

  1. Figure 1 could be improved with additional information. I would have included maybe a few bullet points or keywords explaning the major functional effects of each signaling pathway (from the text).

Response: We appreciate this valuable suggestion. Figure 1 has been revised to improve quality and clarity, incorporating Reviewer #2’s feedback by emphasizing the key functional effects of these interacting pathways in the legend (yellow-highlighted).

  1. Conceptionally, lots of information of given about the effects of Sall4 overexpression in cell cultures, e.g. fibroblasts etc. As a reader I would be more interested to know about the effects of Sall4 overexpression in mice. At the end, lines 460-468 some of this imformation is mentioned, also mentioning Sall4a and Sall4b. Are these twoisoforms of Sall4?? This should be clearly explained. What is the difference between them. Does the unpublished knockin mouse cause overexpression of Sall4a or Sall4b in all or paticular tissues?

Response: We thank the reviewer for this critical observation. The distinction between Sall4a and Sall4b isoforms has been clearly explained in the revised text, along with additional details about the unpublished knock-in model. Please see the updated text highlighted in lines [485–490].

Reviewer 2 Report

Comments and Suggestions for Authors

Author Response

Reviewer 2

The manuscript entitled “Emerging Insights into Sall4’s Role in Cardiac Regenerative Medicine provides a comprehensive and well-written review describing the mechanisms by which Sall4 contributes to the maintenance of stemness, cardiovascular development, cardiomyocyte proliferation and regeneration.

Specific Comments: The inclusion of a Table in subsection 4 entitled Sall4’s Role in Cardiac Cellular Reprogramming and Therapeutics would be beneficial. The information in tabular form should summarize the studies demonstrating the effect of Sall4 on cardiac cellular reprogramming not only in rodents but should also include additional detail defining studies done to address the role of SALL4 in reprogramming human cardiac fibroblasts.

Response: We agree that a table would significantly enhance this section. As suggested, we have included a new table titled "Table 1: Studies Exploring Sall4's Role in Cardiac Reprogramming and Its Human Implications" in the revised manuscript (see page 4 and the yellow-highlighted text).

The format and overall presentation of Figures 1 and 2 could be further improved. A summary Figure that graphically depicts the key steps during which the contribution of Sall4 has the potential to play a key role in Cardiovascular Regenerative Medicine would be useful in Section 6.

Response: We appreciate this valuable suggestion. Figures 1 and 2, along with their legends (highlighted in yellow), have been revised to improve quality, content, and ensure consistency in formatting (see pages 9 and 11). While we agree that a summary figure in Section 6 could enhance the impact, the currently available in vivo studies remain preliminary. Additionally, the complexity of the processes involved in Sall4’s contribution to cardiovascular regeneration may make it challenging to represent accurately in a single summary figure at this stage. Therefore, to maintain scientific accuracy and avoid speculative representation, we have decided not to include an additional figure at this time.